# Information, Thermodynamics and Life: A Narrative Review

George I. Lambrou [1,2,*] , Apostolos Zaravinos [3,*] , Penelope Ioannidou [2] and Dimitrios Koutsouris [2,*]

1    Choremeio Research Laboratory, First Department of Pediatrics, National and Kapodistrian University of Athens, Thivon & Levadeias 8, Goudi, 11527 Athens, Greece
2    Biomedical Engineering Laboratory, School of Electrical and Computer Engineering, National Technical University of Athens, Heroon Polytechneiou 9, Zografou, 15780 Athens, Greece; pioannidou@biomed.ntua.gr
3    Department of Basic Medical Sciences, College of Medicine, Member of QU Health, Qatar University, Doha P.O. Box 2713, Qatar
*    Correspondence: glamprou@med.uoa.gr (G.I.L.); azaravinos@qu.edu.qa (A.Z.); dkoutsou@biomed.ntua.gr (D.K.); Tel.: +30-210-7467427 (G.I.L.); +974-4403-7819 (A.Z.); +30-210-7467427 (D.K.)

**Featured Application: The understanding of the connection between information and life could be proven crucial in our understanding of the brain and how it functions. Information processing takes place probably on all levels of living processes, that is, from the molecular to the macroscopical levels. Therefore, we have attempted to present some considerations and controversies on the topics of information and life along with some applications on how this can be applied to brain-related or brain-similar systems.**

**Abstract:** Information is probably one of the most difficult physical quantities to comprehend. This applies not only to the very definition of information, but also to the physical entity of information, meaning how can it be quantified and measured. In recent years, information theory and its function in systems has been an intense field of study, due to the large increase of available information technology, where the notion of *bit* dominated the information discipline. Information theory also expanded from the "simple" "bit" to the quantal "qubit", which added more variables for consideration. One of the main applications of information theory could be considered the field of "autonomy", which is the main characteristic of living organisms in nature since they all have self-sustainability, motion and self-protection. These traits, along with the ability to be aware of existence, make it difficult and complex to simulate in artificial constructs. There are many approaches to the concept of simulating autonomous behavior, yet there is no conclusive approach to a definite solution to this problem. Recent experimental results have shown that the interaction between machines and neural cells is possible and it consists of a significant tool for the study of complex systems. The present work tries to review the question on the interactions between information and life. It attempts to build a connection between information and thermodynamics in terms of energy consumption and work production, as well as present some possible applications of these physical quantities.

**Keywords:** bit; information; thermodynamic equilibrium; cancer; warburg effect; intelligence; neuro-autonomous systems

## 1. Introduction

Natural sciences are based on the principle of observation and description. In particular, this does not only rely on the motif of observation/description but also on the attempt to describe the phenomenon with a generalized law. Laws (or theories) have the property of being able to predict a phenomenon; that is, based on the present state it is possible to define, within an amount of certainty, the future conditions [1]. On the other hand, biological systems are very complicated to describe and thus it is very difficult to pose generalized theories or mathematical formulations. A possible explanation for the diverse behavior of similar biological systems is that they receive different information from their

surroundings (there is no better way to state this observation than through the words of *Erwin Schrödinger*, who dealt with this subject in his work "*What is life? The Physical Aspect of the Living Cell*". In his own words: " ... *The reason for this was not that the subject was simple enough to be explained without mathematics, but rather that it was much too involved to be fully accessible to mathematics. Another feature which at least induced a semblance of popularity was the lecturer's intention to make clear the fundamental idea, which hovers between biology and physics, to both the physicist and the biologist. For actually, in spite of the variety of topics involved, the whole enterprise is intended to convey one idea only—one small comment on a large and important question. In order not to lose our way, it may be useful to outline the plan very briefly in advance. The large and important and very much discussed question is: how can the events in space and time which take place within the spatial boundary of a living organism be accounted for by physics and chemistry? The preliminary answer which this little book will endeavor to expound and establish can be summarized as follows: the obvious inability of present-day physics and chemistry to account for such events is no reason at all for doubting that they can be accounted for by those sciences* ... " In the same work, *Schrödinger* continues "*Was it absolutely essential for the biological question to dig up the deepest roots and found the picture on quantum mechanics? The conjecture that a gene is a molecule is today, I dare say, a commonplace. Few biologists, whether familiar with quantum theory or not, would disagree with it. On p. 47 we ventured to put it into the mouth of a pre-quantum physicist, as the only reasonable explanation of the observed permanence. The subsequent considerations about isomerism, threshold energy, the paramount role of the ratio W:kT in determining the probability of an isomeric transition—all that could very well be introduced to our purely empirical basis, at any rate without drawing on quantum theory. Why did I so strongly insist on the quantum-mechanical periods for the point of view, though I could not really make it clear in this little book and may well have bored many a reader? Quantum mechanics is the first theoretical aspect which accounts from first principles for all kinds of aggregates of atoms actually encountered in Nature* ... ") [2]. Received information procures changes in all systemic levels of the biological system; that is, on the sub-molecular, molecular, cellular, tissue and organism levels.

Therefore, as anticipated, biological sciences are considered as "accidental sciences" in contrast to the most "classical" natural sciences such as mathematics, physics and chemistry [3]. What is actually suggested is that the "special features of biology as a field are apparent rather than actual because rather than being accidental, biological phenomena are more likely subject to informational rather than physical laws" [3]. Thus, this is in agreement with our opening clause suggesting that "biology is ultimately markedly different from physics, insofar as we understand physics as having laws" [3].

On one hand, current technologies allow the investigation of biological systems down to the level of molecular visualization as e.g., proteins, and on the other hand allow the immense investigation of expression parameters such as genes, proteins, etc. The potential currently provided by high-throughput methodologies of biological systems exceeds our capacity in understanding these data. This fact has always been the main problem in biology, since most descriptions of its phenomena can, and are, almost exclusively contemplative. Thus, prediction of the future states of a biological system still remains elusive. To make this a bit plainer, we will give a very simple example. Suppose we study the presence of a protein in a cell type, under a given condition (e.g., the effect of a growth factor in time). Suppose also that we can detect the protein and measure its levels ($X$) at a given time ($t$), that is, we know $X_t$. Our question is whether we can accurately predict protein levels at time $t + 1$, i.e., $X_{t+1}$. The answer to this question is no, since the only way to do this is to approximate. That is, we can calculate the probability that $X_{t+1}$ takes certain values. Following the previous example, we could pose the same problem for a cell population. Suppose also that we can know the cell population of a system ($N$) at a given time ($t$), i.e., we know $N_t$. Our question is whether we can accurately predict the cell population at time $t + 1$, i.e., $N_{t+1}$. The answer to this question is also elusive, since we are not in a position to predict the cell population in the future.

The aforementioned examples show that there is no way of describing a biological system, just as we do with natural systems. This is the reason why a cellular system will behave differently in Athens as compared to a cellular system in New York. The differences will not be drastic, but if we place both systems at the same time and under the same initial conditions we will not have the same proliferation. The reason for this is most likely, that the two systems during their progression, i.e., their proliferation, will receive different stimuli, i.e., *information*, from the environment and will respond dynamically. It is this dynamic reaction in biological systems that leads them to exchange information and energy with their environment and, therefore, guides their *trajectory*.

### 1.1. The Basic Principles of Information

According to Aristotle's book *Physica*, the purpose of natural science is the explanation of natural phenomena and the investigation of their etiology. In the 20th century, Bohr described the understanding of physics based on empirical evidence from observation and experimentation. Observation and further description is the principal motif of science, leading to the stipulation of laws or theorems, for the prediction of a phenomenon. Moreover, based on *Bohr's* principle of complementarity, it is known that items could be analyzed in terms of contradictory properties (for an example: light behaving as a wave or a stream of particles), and investigates the causality of inspected phenomena. In that way, the ability to predict a phenomenon, i.e., determine its present and future conditions/positions based on the previous and present ones, remains elusive. In order to understand this statement, we could reformulate Galileo's experiment; a rock will fall in the same way if it falls from heights across different geographic locations of the Earth (in particular, the speed of an object falling from a height $h$, is given by the formula $u = \sqrt{2gh}$, where $u$ is the object's speed before it touches the ground, $g$ is the gravity acceleration and $h$ is he height from which the object is falling). If the height is given, we can easily predict the object's trajectory of fall, its velocity and acceleration [1]. However, the same does not apply for the same cell line growing under the same initial conditions in two different parts. This is due to differences in the surroundings. Therefore, biological systems are complex and, thus, it is challenging to form generalized theories or mathematical formulations for them. We can thus attribute the complexity of biological systems in the receipt of different *information* from their environment [4].

In the mid 1940s, Wiener (1948), was one of the first authors to describe information as "information is neither matter nor energy" [5]. Later on, several works have emphasized the role of information in diverse disciplines, such as molecular biology, economics, linguistics and chemical kinetics [6]. Another definition, proposed for information, came from Ashby (1957), who referred to information as the measure of variety in a system indicating the distinct elements in it [7]. Erwin Schrödinger in his book *What is life? The Physical Aspect of the Living Cell* tried to answer some of the basic questions about life. More accurately, he attempted to reason with the matter without the use of mathematics, but with the logic of a physicist. The main objective of his book is not only the different topics in life, but mostly to formulate one idea for the important questions about living organisms. Schrödinger questions: "*How can the events in space and time, which take place within the spatial boundary of a living organism, be accounted for by physics and chemistry?*" In this work, he attempts to combine the essential biological questions with the basics of quantum mechanics and concludes that, as the gene is the core element for the creation of a human being, so the molecule is for matter. This observation is actually the main key of his whole theory that he develops in order to outline the significance of quantum mechanics, because from his point of view science passes from merely observation to the essence of his work. Quantum mechanics is the first theoretical aspect which accounts from first principles for all kinds of aggregates of atoms actually encountered in Nature . . . " [2].

Information in its broader sense has several levels of organization. The first level concerns the data, which are defined as the primary data, the observations and measurements produced by a system and are intended for analysis and processing, with the aim to be



converted to useful information for decision-making, or to draw useful conclusions [8]. Alternatively, data could be considered as a set of objective events, observations, or activities that are recorded and stored, but not organized in such a way that they acquire some particular significance. Data can be in various forms. They can be numerical, alphanumeric, shapes, images, sounds, etc. Examples of such records are the temperature changes in one year, the meteorological observations of an area for a certain period of time (date, temperature, humidity, sunshine, etc.), the altitude of a geographical area, the number of users who wrote the word "flu" in social media, etc. Within a system, data can be inserted through paper or electronic form, language, image or audio, stored on various storage media and produce within a system a sufficient bulk able to be converted into information. Although raw data are not particularly important, they have the characteristic that their collection is relatively simple and can be easily transmitted and stored in an electronic database. When data acquire a measuring unit, it is then that they become useful information. Finally, information can be processed in order to become knowledge, which is the level that living organisms use [9].

Information theory focuses on the valuation (quantification), storage and correspondence (communication) of any sort of data. Shannon in 1948, in his landmark paper "A Mathematical Theory of Communication" [10], proposes a theory about information that includes a transmitter and a receiver for the encoding of every information-message. In this article, the basic elements of communication are described according to the general Shannon diagram and have five key points: (a) an information source, that produces a message; (b) a transmitter, that has the ability to recreate a message as a signal; (c) the signal is sent through the channel; (d) the receiver, that mainly transforms the signal he receives, from the channel into the message that was intended to transfer, and (e) finally the destination which can be a person or a machine for whom the primary message was intended (Figure 1).

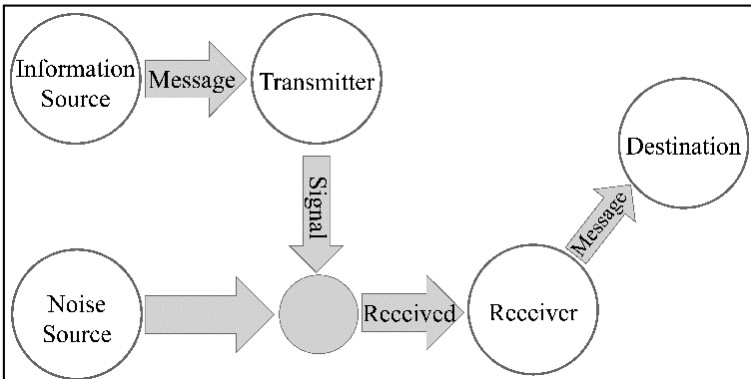

**Figure 1.** Diagrammatical representation of information theory.

### 1.2. Shannon's Information

Information is probably one of the most complex scientific aspects. In 1948 Shannon published a study entitled "*A Mathematical Theory of Communication*" [11], which focused on the various ways of information encoding from a transmitter to a receiver. The main idea of this seminal work was to link information with probability. In this fundamental work, Shannon mainly used the concept of probability theory in information. If $Z^L$ is the set of all "words" of a length $L$ for a finite alphabet $Z$, then each "word" $w$ is a possible message sent by an information source, and consists of a "stationary stochastic process" [12]. Thus, if $P(w)$ is defined as the probability for a word w to be sent as a message the information it contains under Shannon's information definition is:

$$I(w) := -\log_2 P(w) \tag{1}$$

In that sense Shannon introduced the term of "informational source" as a pair *(X,p)*, where *X* is a finite set of objects and *p* is the probability function assigning to every $x \in X$ the probability *p(x)* of occurrence. This approach had as a consequence the definition of the average quantity of an information source *(X,p)*, which is defined as:

$$H(X, p) = -\sum_{x \in X} p(x) \log p(x) \qquad (2)$$

The *H(X,p)* function is also called the entropy function, where the total entropy can be described:

$$H_j = \sum_{i=0}^{n} H_{j,i} \qquad (3)$$

The first applications of his theory included the computation of Channel Capacity with the induction of information to *bits*, which is the abbreviation for "*binary digit*" [9]. Bits can exist in two states, which can be abstractly represented by the digits 0 and 1. In that sense every information can be described as the combination of binaries.

Further on, bits can be used to implement information by means of a semiconductor, magnets, condensers and so on. In general, information can be divided in three main processes: (a) Storage, (b) Transmittance and (c) Processing [10,11]. Moreover, information should have a content, some sort of meaning that will make sense to the receiver (semantics) [13] and finally, a critical aspect of information theory is the transmission. This includes several discrete stages, which are: (a) the message, which includes the information per se, (b) the transmitter, (c) the encoder, which transforms the information into bits, (d) the channel of information transmittance, (e) the decoder, which decodes the digital information, (f) the receiver (also called the destination) and (g) the noise source, which is beyond the control of the transmittance process and interfered with the information (Table 1).

**Table 1.** The process of transmission plays an important role in establishing the terms of information theory. The present table shows the key elements involved.

| | Stage | Description |
|---|---|---|
| 1 | *The message* | Includes the information itself |
| 2 | *The transmitter (source)* | It is the sender of the message |
| 3 | *The encoder* | The encoder represents the message in a sequence of bits (or other symbols) based on a rule |
| 4 | *The channel* | is the conduit for information |
| 5 | *The decoder* | reverses the process of encoding the message and represents the message in a format understandable to the recipient |
| 6 | *The receiver (destination)* | It is the recipient and the signifier of the message |
| 7 | *The noise source* | It is the environment of the system and is a factor that is beyond our control. It is a cause of information degradation. We believe that it interferes with the information as it spreads through the transmission channel. |

Entropy in Shannon's Information

The most basic concept of classical information theory is that of Shannon entropy [11,13]. This quantity is related to the information load of measuring the value of a random variable *X*. In a sense, entropy measures the uncertainty we have about predicting the value of *X* before observation. Thus, in the literature entropy is also referred to as an uncertainty measure. The importance of entropy as an information load is fundamental to applications because its value determines the minimum amount of information we need to retain from a message in order to be able to reproduce the original information. This result is included in Shannon's Noiseless Coding Theorem [14,15], which is fundamental to information theory. Essentially, the concept of entropy comes to answer the basic question that has to do with how much natural resources we need to consume in order to manage some information.

The importance of entropy as an information load is fundamental to applications because its value determines the minimum amount of information we need to retain from a message in order to be able to reproduce the original information. This result is included in Shannon's Noiseless Coding Theorem [14,15], which is fundamental to information theory. Essentially, the concept of entropy comes to answer the basic question that has to do with how much natural resources we should consume in order to manage some information [9].

### 1.3. Quantum Information

In 1989, John Archibald Wheeler tried to unify the concept of information with physics and quantum theory [16]. Its basic definition was " . . . every physical quantity, every '***it***', derives its ultimate significance from bits, binary yes-or-no indications . . . ". This statement was summarized in the phrase "'***it***' from '***bit***'". Wheeler's study included evidence from Bohr's position on information, who argued that quantum mechanics and relativity lead us to abandon the shackles of the world's visual perception, and that what seemed particularly important was language (linguistics). In simpler terms, Bohr said that regardless of any physical condition, people need a language to communicate, any form of language either vocal or visual [17]. As aforementioned, language (more generally communication) is the toolbox through which the experience is perceived as well as its analysis. Beyond that, however, Wheeler described bits as the "quantum of reality", in his own words " . . . *I suggest that we may never understand this strange thing, the quantum, until we understand how information may underlie reality. Information may not be just what we 'learn' about the world. It may be what 'makes' the world. An example of the idea of 'it' from 'bit': when a photon is absorbed, and thereby 'measured'—until its absorption, it had no true reality—an unsplittable bit of information is added to what we know about the world*" and " . . . *at the same time, that bit of information determines the structure of one small part of the world. It 'creates' the reality of the time and place of that photon's interaction* . . . " [18].

Quantum information matches digital information and bit, where the unit of measurement is the "*qubit*". Quantum information differs from classical information in many critical aspects. In digital information, the values that a system can take are two and are distinct, *0* and *1*. In the case of quantum information, the "*qubit*" and is a continuous variable, described by the direction of a vector in a sphere termed as the Bloch's sphere (Figure 2).

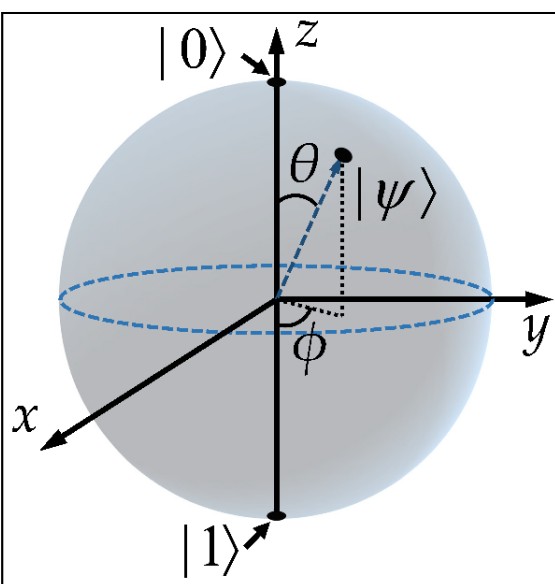

**Figure 2.** Bloch's sphere is a geometrical representation of a two-level quantum mechanical system, the qubit. The vector $|\psi\rangle$ takes values between $|0\rangle$ and $|1\rangle$.

As qubit's nature is the elementary unit of information, the Heisenberg Uncertainty Principle applies ("In quantum mechanics, the uncertainty principle (also known as Heisenberg's uncertainty principle) is any of a variety of mathematical inequalities asserting a fundamental limit to the precision with which the values for certain pairs of physical quantities of a particle, such as position, *x*, and momentum, *p*, can be predicted from initial conditions. Such variable pairs are known as complementary variables or canonically conjugate variables and, depending on interpretation, the uncertainty principle limits to what extent such conjugate properties maintain their approximate meaning, as the mathematical framework of quantum physics does not support the notion of simultaneously well-defined conjugate properties expressed by a single value. The uncertainty principle implies that it is in general not possible to predict the value of a quantity with arbitrary certainty, even if all initial conditions are specified" (from https://en.wikipedia.org/wiki/Uncertainty_principle#cite_note-Sen2014-1, accessed on 16 October 2020).) [19]. Thus, it is impossible to measure the value of the elementary unit accurately. In addition, a qubit cannot be translated into bits, due to the non-teleportation theorem ("In quantum information theory, the no-teleportation theorem states that an arbitrary quantum state cannot be converted into a sequence of classical bits (or even an infinite number of such bits); nor can such bits be used to reconstruct the original state, thus 'teleporting' it by merely moving classical bits around. Put another way, it states that the unit of quantum information, the qubit, cannot be exactly, precisely converted into classical information bits. In crude terms, the no-teleportation theorem stems from the Heisenberg uncertainty principle and the *Einstein–Podolsky–Rosen* paradox: although a qubit can be imagined to be a specific direction on the Bloch sphere, that direction cannot be measured precisely; if it could, the results of that measurement would be describable with words, i.e., classical information. If it were possible to convert a qubit into classical bits, then a qubit would be easy to copy (since classical bits are trivially copy-able)" (from https://en.wikipedia.org/wiki/No-teleportation_theorem, accessed 20 October 2020).). Although this theorem holds, it is possible to move qubits from one physical entity to another via quantum teleportation. The qubit cannot be copied or deleted. It cannot be delivered to more than one recipient. A qubit can be changed by applying linear transformations and/or quantum gates to it, the analog of logical circuits in digital information. Qubits can be synthesized and result in digital sets. Quantum information can be transferred to a communication channel, like the classic example of communication channels. The simplest quantum system is described in a complex two-dimensional vector space (2-D Hilbert space C2) [9].

If we define an orthogonal reference system in a Hilbert space and let $|0\rangle$ and $|1\rangle$ be two constant vectors, then any state of the system can be expressed as a linear combination of these vectors such as $|\Psi\rangle = \alpha|0\rangle + \beta|1\rangle$, where $\alpha$ and $\beta$ are complex numbers and must satisfy the condition $|\alpha^2| + |\beta^2| = 1$. If those conditions hold then $|\psi\rangle$ is said to be normalized. The conditions $|0\rangle$ and $|1\rangle$ are also referred to as computational basis states.

It is useful to make a comparative presentation of the quantum bit with the classic bit. The bit can only exist in one of the discrete states 0 and 1 at a time. In contrast, qubit has a continuous state space and can exist in any state that can be described by the complex numbers *a* and *b*. In the case of bit we can always be sure if our system is in state 0 or 1. On the contrary, for a qubit we cannot say with certainty which of the two states it is in. For example, if we have only one qubit, we can in no way make a measurement that will tell us if our system was in state 0 or the state before the measurement [9]. Determining the initial state, however, it is statistically possible if we have a large number of similar initial states. The difference between qubits and classic bits becomes more apparent when we have to compare systems with more bits. Two classic bits can be found in four different states: 00, 01, 10 and 11. In the case of two qubits, any linear combination of the four base vectors is possible. This means that the two qubits can be found even in states of complete quantum correlation or otherwise in states that are entangled. Such situations are all four Bell states: qubits are useful in the theoretical understanding of the meaning of information, in finding

practical methods of its representation and in its processing. Therefore, the elementary unit of quantum information should be able to change its state and is adapted based on specific mechanisms so that it can store and process information. The mathematical representation of these mechanisms of change (transformations) is done with the help of linear operators. The four basic transformations are the following (Pauli operators): The transformations of one qubit or more are called quantum gates in the language of information theory in correspondence with the logic gates of classical theory. Quantum gates are generally unit transformations. Pauli transformations are an example of quantum gates. The *X* operator is often called the quantum not gate because it overlays the computer base vectors and. Operator *Z* is also known as phase flip gate because it holds that and. There are several reasons why we take qubit as the fundamental amount of quantum information. First of all, the qubit is the simplest quantum. Also, any finite dimensional quantum system can be described by an equivalent system consisting of a finite number of qubits. Of course, other basic information storage systems have been devised, such as qutrits, similar to classic trits (three-state systems), but they can again be defined as a subsystem of qubits. For these reasons qubit has prevailed as the fundamental unit of quantum information. All of the above are differences between the two types of information, digital/classical and quantum.

### 1.4. Information and Thermodynamics

At this point another question arises and that is what is the connection between information and thermodynamics (the father of thermodynamics is considered to be the French physicist Nicolas Leonard Carnot (1 June 1796–24 August 1832†)); This question has been faced by Landauer, who states that "*in order to completely delete an information, energy must be consumed*" [20]. However, it is difficult to measure information directly and the proposed way is to define it through "ignorance" that is missing information. As previously stated, it is possible to define "ignorance" mathematically and thus information can be defined based on the difference of the level of "ignorance" before and after receiving information [6]. However, progress in this area and how they relate to biological phenomena can be attributed to the two aforementioned major scientists: Max Karl Ernst Ludwig Planck (23 April 1858–4 October 1947), father of quantum theory, and Erwin Rudolf Josef Alexander Schrödinger (12 August 1887–4 January 1961†) father of quantum mechanics. With their work, the foundations for the coupling of biology and physics were essentially laid. In fact, for the first time, the relationship between thermodynamics and biological systems is discussed in such a way as to show that there is a clear connection which, however, was not and is not capable of being described with the existing mathematical formalities.

To better understand the aforementioned concepts, it is interesting to examine the relationship between thermodynamic equilibrium and cell proliferation and, therefore, proliferative dynamics. As is well known, biological systems receive stimuli from the environment, that is, information, and exchange energy and matter. We have already mentioned this before, and here we further specify it with reference to competition for energy resources. It is also known that biological systems are open, that is, they exchange energy and mass with their environment. Another fact that we know for sure is that they operate out of thermodynamic equilibrium. To put it simply, the cells of eukaryotic organisms operate at a lower ambient temperature (e.g., on a cool spring night), but often much higher (e.g., in a sauna). In both cases there is a way for the cells to maintain a constant body temperature and not equilibrate with their environment. This is an example of an operation out of thermodynamic equilibrium. Another example is the function of reaction enzymes as catalysts, which would not be possible in any other case. So, we could say that biological systems have three properties: they exchange energy and mass with their environment, they compete for resources (mass, space and energy) and, therefore, they multiply and in fact not erratically but in a coordinated way. These three phenomena may be formulated as a theorem but will not be the subject of the present work.

## 2. Information and Life

All living organisms have three basic properties: (a) they have a structure, which isolates them from the environment (e.g., the cell membrane), (b) they maintain a minimum of entropy (i.e., they operate out of thermodynamic equilibrium) [21] and (c) (and most importantly) they are compatible with life. We emphasize point (c) in particular because even the pathology in an organism, caused by invaders such as bacteria or by cells of the same organism, such as neoplasms, is compatible with life. Living organisms have two more points that make them stand out. They can reproduce and receive information from the environment, which they store in the form of a genome. In a recent work it was suggested that the concept of information at the biological level is intertwined with the ability to produce work (from biological systems) [22]. Although it is quite clear that the three elements of information, namely transmission, processing and storage are key components for the preservation of life, the basic mechanisms and the role of information, remain unclear and subject of intensive study. This problem becomes even clearer by the fact that we cannot give an accurate and unique definition of biological information as well as its measurement. Many times, in order to bypass this "reef", biological information is defined as a quantity, which is entailed within the biological system and its dynamics. In fact, this is considered to be true to such an extent that it is impossible to separate the two entities, the biological system (chemistry) and information [22].

Of particular interest is the fact that it is difficult to distinguish between what is information and what is not. Let us consider for example a closed cell system in vitro. The cells are in a culture medium, which provides nutrients and constant pH, and an incubator provides a constant temperature and oxygen. Assuming that cells are allowed to proliferate without any restrictions and constant nutrients concentrations, then this means that within a given period of time they will starve. Absence of nutrients is an environmental stimulus, which leads to a number of reaction mechanisms by the cell, which try to maintain their homeostasis and life. At the cellular and molecular levels, both genomic and non-genomic responses are likely to consist of information flowing from one direction to another. The question, however, is whether the original condition of starvation is in turn information.

### 2.1. The Old Problem of Maxwell's "Demon"

To better understand the meaning of information, we need to refer to Maxwell's Demon. This is an imaginary experiment devised by James Clerk Maxwell and connects the concepts of information and entropy with energy. This hypothetical experiment was designed to better understand and possibly 'overturn' the second law of thermodynamics which forbids the production of energy from scratch in a closed system. Maxwell first published his conception of this imaginary experiment in 1871 in his book *Theory of Heat* [23], in a section on the limits of the second law of thermodynamics. Maxwell's term, 'demon', is attributed to William Thompson. In the device of the experiment, a "being" capable of knowing at any time the speed and position of each molecule of a gas, in a container divided into two parts, pulls and opens a door at will and allows the "cold" (low speed) and the "hot" (high speed) molecules to separate. The door is allowed to close immediately after the selective passage of each molecule and the spring returns to the demon the energy it expended to open it. In the end, without giving energy to the system, which is considered isolated from the environment, one side of the box appears with high-speed particles, thus hot gas, and the other with low-speed particles, thus cold gas, (and possibly different pressure in one container from the other). This leads to an increase in the energy of the system, since we can combine the two heat tanks (left and right), with different temperatures, with a heat engine in order to produce work (Figure 3).

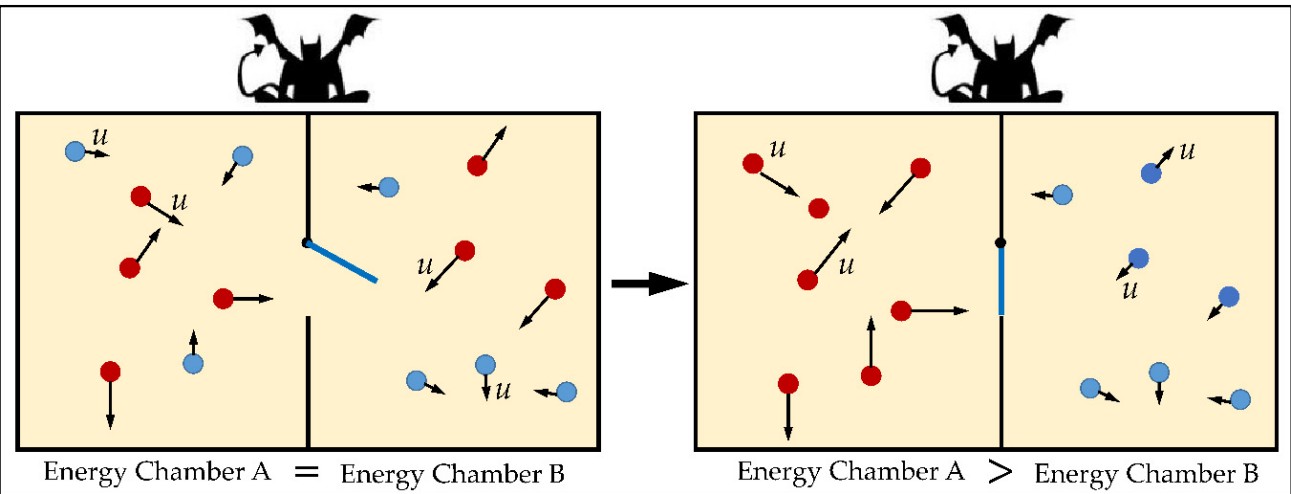

**Figure 3.** Maxwell's Demon. High-speed, thus high-energy, particles (red circles) and low-speed, thus low-energy, particles (blue circles) co-exist in two chambers in equilibrium, meaning that the total energy is the same in both chambers. A demon can open the door in the center of the box and, therefore, separate the high-speed from the low-speed particles. After a while, the demon would have formed a new situation, where the high-speed particles would be prevalent in one chamber and the low-speed particles in the other. That way a box would have formed far from equilibrium, and in particular one chamber will be hotter than the other.

However, the existence of a "being" with such characteristics, capable of performing all this information processing without wasting energy, at least as much as it produces by separating particles to hot and cold, proves impossible. To transform a disordered system (mixed molecules) to an ordered system (the molecules are organized spatially according to their velocities), that is, to obtain information (which describes a more organized system) from the processing of the data of each molecule, requires energy expenditure. The increase in information, resulting from the description of a more organized system, is equivalent to a decrease in entropy, which is prohibited by the second law of thermodynamics for a closed system. This experiment was given many interpretations and finally it turned out that the second law of thermodynamics applies. To be more precise, the demon must waste energy to monitor and record all their molecules and velocities and decide which is slow and which is fast as they are constantly exchanging energy with each other. It needs to constantly send messages to each molecule in order to return the coded information of their position and speed, or to be able with some series of measurements to be able to receive this information. This information must be stored and processed in order to decide when to open and close the door, for example it must be careful not to be a "fast" molecule near the door and in the direction of the "cold" container when it opens.

One of the most famous answers to the problem was proposed in 1929 by Leó Szilárd [24] and later by Léon Brillouin [25]. According to their interpretation, since the demon and the particles interact, we must consider that the total entropy of the system is the combination of the entropy of both. The increase in the entropy of the demon from the measurement process would ultimately be greater than the decrease in the entropy of the particles, so that the total entropy would increase. Assuming that information transfer by molecules was performed by a reversible thermodynamic processes, Rolf Landauer argued in 1960 that the measurement process was possible without increasing entropy [20]. This reasoning would apply, provided that the information collected and stored is not erased, as any reduction in information equates to an increase in entropy. The final proof came in 1982 from Charles H. Bennett. Bennett showed that no matter how well prepared the demon was, he would eventually have no memory available to store and process the information he would receive, and would have to begin erasing some of the information he had previously collected to reuse the memory. Deleting information, however, is an irreversible

process, which means that eventually the entropy would increase, that is, we would not be violating the second law of thermodynamics and producing energy from scratch.

### 2.2. Information in the Cellular Context

Based on these observations, we could postulate the following: to reduce entropy requires knowledge, i.e., information. Here, however, things are becoming more complicated since we mentioned two concepts, knowledge and information. It is not enough for the Demon only to know that there are molecules present in the box, but he also must have knowledge of their properties. Similarly, a random series of letters on a board, could be information for an observing party, yet putting them in the correct order to form words is knowledge, that is, useful information.

Similarly in a cellular system, if a random amino acid sequence is produced in the ribosomes, they will take such conformation in order to obtain a minimum energy level, however the amino acid sequence will bear no functional role in the cell. Therefore, the appropriate amino acid sequence must be formed. Perhaps, drawing a parallel between the Demon and the cell, we could say that the cell sorts the erratically "spoken" information of the environment, since the variety of stimuli can be infinite, and responds in such a way as to maintain its homeostasis, spending energy. This principle, in fact, has been reported to be applicable as a generalized theory of planet Earth [26].

In addition, let us consider a more complex issue in biology, which is cancer. To date, it is not entirely clear why it appears and there are many theories about the mechanisms of carcinogenesis. In a recent study it was stated that carcinogenesis is directly related to thermodynamics and entropy [21]. The basic premise is that biological systems, in order to be thermodynamically stable (i.e., to function), must have the maximum (?) amount of information. For example, from prokaryotes to eukaryotes, information maximized as mitochondria were added to the second, providing energy and autonomy. Therefore, the gradual transition from the healthy cell to the cancer cell represents the reverse process, where information is minimized, that is, we move on to what is called *information destruction*. This process takes place in discrete stages. The first is genomic instability, or genomic abnormalities. Accumulation of chromosomal and sequential mutations probably leading to neoplasms. The second is cellular instability, where the cancer cell differs both morphologically and phenotypically from the normal cell and finally the inability to "process time". Cancer cells are virtually immortal. In fact, it is believed that if they did not lead their host to death, they could live forever [21].

At this point, however, some discrepancies arise. The above formulation for the nature of cancer cells is directly related to our perception of the tumor cell. In the clinical setting, tumor cells are considered as a pathophysiological condition, where its presence is incompatible with life and therefore justifies the view of minimizing information. There is another view, however, which sees the tumor cell as an evolution of the normal cell. After certain stimuli, the cell comes to a "crossroad", in which it has to choose between apoptosis and immortality, and in the case of tumors it chooses immortality. We would even dare to say that the cell passes from the "mortal" state that possesses and passes to another "higher" state, which is "immortality". In addition, the progression of a tumor, after its appearance, signifies anything but lack of information. The homeostasis mechanisms, the tumor cell acquires, are so powerful that it is able to overcome any attempt to eliminate it by becoming resistant to chemotherapeutic agents even in radiation, both extremely lethal to normal cells. Also, the fact of genomic abnormality, which is observed in cancer cells cannot be considered absolutely as the general causative factor but rather as the result. This is due to the fact that mutations, which are present in a cancer cell, manifest in frequencies (that is a mutation is present in a fraction of tumor cells) and not in their entirety (that is, a mutation is not present in all tumor samples). For example, a well-studied factor is the TP53 protein, for which dozens of mutations have been found to date, but which are also found in normal cells, without being a sufficient and necessary condition for carcinogenesis.

One last point, and perhaps one of the most interesting, is the energy balance in cancer. One of the first theories of carcinogenesis was formulated by Otto Warburg, who observed that mitochondria malfunctioned in cancer cells [27]. Up-to-date it is known that the energy needs of the cancer cell are not covered through oxidative phosphorylation, which has the highest efficiency in the oxidation of glucose but through glycolysis, which has a very low energy efficiency and a significant by-product, the lactic acid. Therefore, it has been suggested that carcinogenesis could be parallelized to the loss of energy efficiency in a stem cell [21]. In other words, it has been suggested that cancer cells cannot maintain high levels of information and, therefore, have energy losses.

However, taking a closer look at the aforementioned statement, it appears that it has an inherent error. In the case of tumors, the energy needs are much greater than those of a normal cell. In fact, we are talking about large amounts of wasted energy. To highlight our concept, let us consider that in oxidative phosphorylation one molecule of glucose produces 12 molecules of ATP while in glycolysis, one molecule of glucose produces 2 molecules of ATP. This means that much larger amounts of energy must be expended (as units of glucose molecules, regardless of their final yield) in order to keep the cancer cell alive. Thus, from this point of view we can really talk about a waste of resources and not about reduction. In order to further determine whether the energy reduction formulation is true and, therefore, to link this to the fact that information retention is intertwined with work production, the work produced by a cancer cell must first be measured, both at rest as well as during division compared to a normal one. This question has not yet been answered and is the subject of extensive research.

### 2.3. The Information Flow

With these thoughts in mind, we come to analyze an interesting part of our topic, as was suggested at the beginning of this paper. That is, what is the thermodynamic bearing of intelligence (from both sides)? Returning to the example of the Demon, we can ask the following question: does it matter, in terms of the energy burden that will be spent, whether the Demon is intelligent or not? In other words, will an intelligent Demon consume more energy to achieve the separation of particles from a less intelligent one, even assuming that they will both follow the same process of measurements and criteria? The question, as far as we know, has not been answered. To complicate the matter a little more, if we consider that the energy required for the birth of a human being is $W$, is there a difference between $W_{More\_Intelligent}$ and $W_{Intelligent}$? At this point we can assume that most likely we do not have any differences in the energy load of the two subjects. Even if one or the other case requires more or less methylation, or different nucleotide sequences, the energy content of the genes and proteins that play a role in it is the same, and if not the same their differences would be negligible. In addition, both subjects, the smart and the less intelligent, as they grow older will consume amounts of energy to grow and, therefore, we can assume that they will spend the same amounts (assuming that both subjects have equal treatment and as well as equal opportunities in education and life). Let us now turn to the subject of information. It is certain that in terms of information, both in terms of transmission, processing and storage, the more intelligent subject will be able to act with greater amounts of information than the intelligent subject and, therefore, based on what we have mentioned, larger amounts of energy will be required (Figure 4).

In Figure 4 we present the transition flow from one state to the next from the stage of conception to the interaction with society. A paradox emerges from this diagram. How is it possible to start from the same energy levels, assuming that the energy costs (food, housing, education etc.) remain constant for both subjects but the effect of information flow, work production, is greater for one subject (the most intelligent) as compared to the intelligent subject? As mentioned, we have not encountered a study that deals with this issue in the literature.

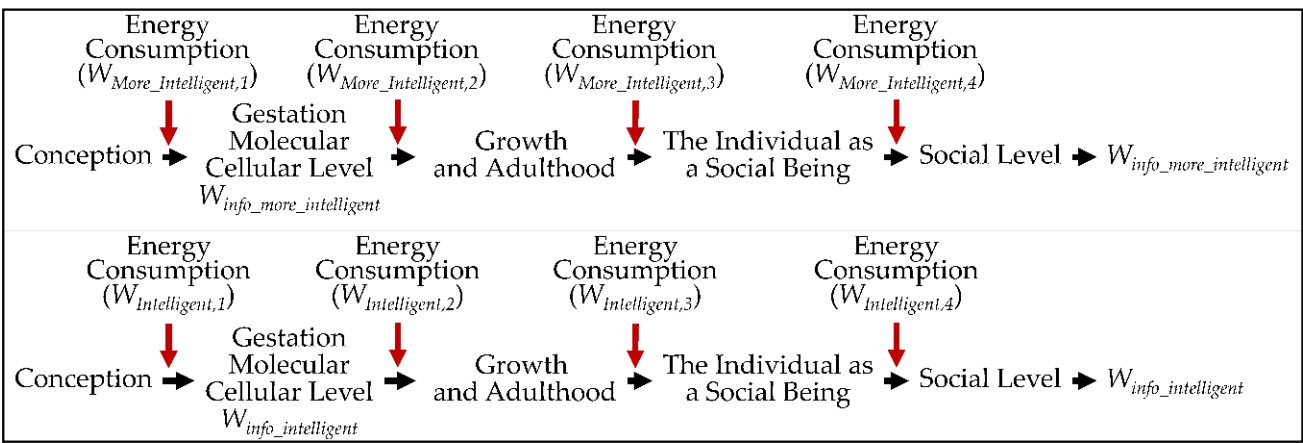

**Figure 4.** Energy flow from conception to social beings. The transition from one stage to the next requires the expenditure of work (energy) yet the work produced at the final stage that is when the social subject is required to produce work, does not equal the total energy expenditure. In other words, $W_{More\_Intelligent,1} + W_{More\_Intelligent,2} + W_{More\_Intelligent,3} + W_{More\_Intelligent,4} \neq W_{Info\_More\_Intelligent}$ and $W_{Intelligent,1} + W_{Intelligent,2} + W_{Intelligent,3} + W_{Intelligent,4} \neq W_{Info\_Intelligent}$.

It is known that the brain consumes the largest amount of glucose as compared to other organs. Thinking is an expensive "sport", but it has not yet been shown that a clever thought can "cost" more energy than a less smart thought. In fact, it could be just the opposite. A more intelligent mind expends less energy than a less intelligent mind to do exactly the same process. We return to the previous paradox. The sum of the work produced, from the moment of pregnancy until the moment of the first transmission of information is $\Sigma W_{More\_Intelligent} \cong \Sigma W_{Intelligent}$. This equality changes with the transmission of information, where produced work increases in direct connection to the information obtained, which in turn produces more work. Yet, it is possible that two different subjects will produce very different levels of work based on their intelligence capacities. Based on these observations, it seems that the thermodynamic balance is disturbed. If we decide that the thermodynamic equilibrium must remain undisturbed there is one and only one theory that could support this. Yet, we still have the problem on how to quantify the effect of information on the intelligence and the subsequent produced work. That means if two subjects have consumed similar, or comparable amounts of energy to reach different intelligence levels, how is it possible that they will produce unequal amounts of work? For the law of thermodynamics to apply, there is only one explanation; information carries energy! This issue has not been studied to date and to the extent that we are able to know. It would be extremely interesting (but also a big bioethical gap) to study this phenomenon, since we would contribute to a better understanding of the natural world that surrounds us. On that topic, an interesting work has investigated the thermodynamic cost of fast thought [28]. According to previous theories, information becomes meaningful if it can be "anchored" to relevant aspects of the preexisting cognitive structure, which are known as the "subsumers". In this context, it is suggested that human cognitive functioning can be divided in two main categories; "intuition" (which it is referred to as "system 1") and "reasoning" (which it is referred to as "system 2"). "Intuition" is thought to be the process of automatic, instinctive thought while "reasoning" is thought to be the voluntary, logically deductive type of thinking (which is actually much slower). Without getting into this theory in detail, it is proposed that the "thoughtful" process of an individual can be described by:

$$\frac{d}{dt}S(s_1, s_1, \ldots, s_N) = \sum_{i,j=1}^{N} D_{S_i I_j} I_j + f(s, \bar{s}) \tag{4}$$

where, $I$ are the units of information that are used as input, $s_i = s(t)$, $S(s_1, s_2, \ldots, s_N)$ are the binding sites of cognitive structure (the "subsumers") and $D_{S_i I_j} I_j$ is the similarity metric

of information (*I*) and the cognitive structures. The $\sum_{i,j=1}^{N} D_{S_i I_j} I_j$ term is the progressive differentiation of the learning process in a "system 1" type. Finally, $D_{S_i I_j} I_j$ is a normalized metric that accounts for the correlation between $s_i = s(t)$ and information *I*. The term $f(s, \bar{s})$ is an "exchange term" representing the sum of the self-interactions between "subsumers", representing the process of integrative reconciliation governed by a "system 2" type. Finally, the rate $\frac{dS}{dt}$ is the modification rate of the cognitive structures with respect to time. If this reasoning could be applied to Maxwell's Demon, then:

$$\frac{dS}{dt} = D_{S_i I_j} I_j \tag{5}$$

where the change rate of cognitive structures, equal to the similarity metric, also called a "basal subsumption process". These formulations were stated in order to separate between fast thinking ("intuition") and slow thinking ("reasoning"). In another seminal work, the "fundamental equation of information science" has been proposed [29]. This statement, was considered to be the transition state of an information (*I*), with the effect of a cognitive framework. In particular, it was defined as:

$$K(S) = I \rightarrow K(S + \Delta S) \tag{6}$$

where, the state of thinking *K(S)* is equal to the information *I* plus a next state *K(S + ΔS)*. Given these formulations for a time-independent input, this equation can be solved as:

$$\int_{\Delta t} I dt = \frac{1}{D_{s,I}} \int_{K(S)}^{K(S+\Delta S)} \frac{dS}{S} \tag{7}$$

where, *K(S)* and *K(S + ΔS)* could be the prior and later states of a Maxwell's demon.

Yet, what is the work produced by the information under "fast" and "slow" thinking? In a previous work, investigating the work from a "demon" it was suggested that the maximum work obtained could described by [30]:

$$W \leq kTI \tag{8}$$

where, *k* is the Boltzmann constant, *T* the temperature and *I* the information quantity exchanged. Further on, it was suggested that the time-dependent solution to the formulation of Equation (8), is defined as [31]:

$$W_{\max}(t) = k \int_0^t TI dt \leq kTI(t) \tag{9}$$

where *k* is the Boltzmann constant, *T* the temperature and *I* the information quantity exchanged in time *t*. Based on the aforementioned equations, the smaller the time interval the smaller the work produced, and vice versa. Yet, this could not always be true. Although, these formulations describe probable differences between "fast" and "slow" thinking, they do not describe possible differences between "smart" and "less smart" thinking, where both can be "fast" or slow", thus time-independent. Despite the intriguing question, to the best of our knowledge there has been no known answer to date.

### 2.4. Information and Genome

Eukaryotic cells have three common characteristics: (a) they are enclosed in a membrane isolated them from their surroundings, (b) they minimize entropy, functioning beyond thermodynamic equilibrium [21], and (c) they are compatible with life. The last trait includes both their well-being, but also their pathological state, such as cancer. Bio-

logical systems are also able to proliferate and exchange matter, energy and information with their surroundings [22]. Despite the fact that the three elements of information, i.e., "transmission, processing and storage" make up an essential portrait for life, from the information point of view, we know that an information transfer takes place in the processes of life, but we do not know how this is achieved. In addition, it becomes more complicated since the very meaning of information, its role and measurement in physical systems is a subject of research. In particular, in the genomic context it is certain that besides the molecular exchange during transcription and translation, the exchange of information takes place, but we do not know how. The content of information in biological systems has been approximated through the "memory" (that is the quantity) that a biological system can store and, thus, procure its dynamic evolution. Further on, a plausible hypothesis would be that in living systems it is probably not possible to separate the chemical "lattice" of life from its informational content [22]. Besides that, we are still not in position to define clearly how information is "saved" and transmitted in the genomic context. Several theories have been advanced, of which some include the conceptualization of the genomic information as a formal natural language [32,33], an algorithm [34], a mathematical model [35], or an informational theory aspect [36].

The information content of a eukaryotic cellular entity consists of the time-dependent accumulation of translated intracellular and acquired extracellular information. This information state controls the morphology and function of that cell, as well as its interaction with the external environment. Anomalies in cellular "*informatics*" can result in disease, as clearly exemplified by several genetic disorders [22,37].

A very elementary trait of biological systems that discriminates them from other physical dynamical systems, is the presence of the genome. In particular, the core informational content is in the DNA, which is transmitted through transcription (RNA) and translation (proteins) [38,39]. At the same time, life could be considered as the "cross-talk of genes" interacting in a complex order, of the storage, processing, and propagation of cellular information maintained in the genome. This general class of phenomenon can be addressed through information theory developed by Shannon [40] and Fisher [41] to measure information content and communication. Their studies have contributed to the physical sciences and biology [22,37].

In a previous work by Johnson, HA (1970), information theory was characterized as the "new calculus" for biology. It is apparent that living systems require imperatively matter and energy i.e., exchanging those with their environment. Moreover, it has been emphasized that "*life without information is likewise impossible*" [42]. Since that time, remarkable progress has been made towards understanding the informational basis for life. High-throughput methodologies allow the complete characterization of the RNA and protein contents of cellular populations and even of individual cells. The human genome (as well as the genome of other species) has been sequenced, and provided an immense amount of information.

Biological systems (or living organisms) are complex systems, operating beyond the thermodynamic equilibrium. They can exchange energy and mass with their environment, and manifest self-sustainability, motion and self-protection at the same time. Another significant trait of biological systems is the presence of a genome, entailing these traits, and the ability of self-awareness of their existence. These traits make it difficult and complex to simulate them in artificial constructs. There are many approaches to simulate autonomous behavior, yet no one is conclusive.

One of the main problems in simulating living systems is the transfer and processing of information. At the basic level information is transferred from DNA to mRNA to protein (with the exception of retroviruses, which transfer information in reverse, from RNA to DNA). The complexity of these mechanisms made their understanding tedious. Until the early 1990s, the main scientific approach was to study the different phenomena (i.e., molecules or cellular events) in a sequential order, meaning that researchers could discover the function and role(s) of biological molecules in small numbers and one step

at a time. Yet, in the early 2000s high-throughput methodologies emerged (microarrays first emerged in 1995), and allowed the study of thousands (or even millions) of factors, simultaneously. Thus, due to the immense availability of data, the need for mathematical modelling emerged.

Attempting to model biological systems is not new. The first approaches appeared in the advent of the 20th century. One of these was named physical biology. As Henri Poincare mentioned: " ... *life is a relationship among molecules and not a property of any molecule ... Science is built up of facts, as a house is with stones. But a collection of facts is no more a science than a heap of stones is a house ...* " [4,43]. Despite the immense availability in biological data, we are still unable to comprehend the exact mechanics of information transition in biological systems.

One of the main difficulties in simulating biological systems is that they are dynamical in nature. That means that they can adapt to random changes in their environment over time. This "obstacle" remains crucial towards the construction of artificial autonomous dynamical systems or hybrids. These observations bring about another issue, which is the differentiation between stochasticity, randomness and "chaoticity". Although what seems a rather trivial observation for living systems to be stochastic in nature, in reality they are chaotic. More specifically, biological systems are the ones that determine their equations of "motion", instead of the classical paradigm where an equation can describe the motion of a system [44]. In the case of dynamical chaos, discovered by Poincare, no sufficient explanation was given why such systems follow infinitely dynamical trajectories or perturbations depending on their initial conditions [45]. To this problem, several approaches have been proposed where one relatively recent proposal was described as the "*approximation-free, coordinate-free supersymmetric theory of stochastic partial differential equations*" [44–47]. This theory, referred to as the supersymmetric theory of stochastic dynamics (STS) [48], entails the set of mathematical models, which present a large applicability covering, dynamical systems, with and without noise. From the physical point of view, this theory describes the ubiquitous and "spontaneous", long-range dynamical behavior of complex physical systems, such as earthquakes, neuro-avalanches, as living systems [44,47].

## 3. Applications "after" Information: To the Process of Learning

Having the information at hand is not necessarily useful or work-producing. Information must be converted to knowledge, through the process of learning. This concept was studied by Bush and Mosteller (1957), where a general theory of learning and stochastic process was proposed [43]. For several years investigators have been developing a mathematical model for describing a variety of experiments on animal and human learning. This model was closely related to that developed by Estes and further by Miller and McGill [49,50]. These models have led quite naturally into a study of a class of stochastic processes, which may be viewed as Markov chains with an infinite number of states. In applying the model to the analysis of experimental data, a number of problems in statistical estimation have risen. A learning process, as the term is used, involves systematic changes in behavior; one type of behavior may become more frequent and another type of behavior may become less so. The authors describe this learning process in a situation where a choice of a number of given alternatives occurs periodically. Each occasion on which there is an opportunity for making a choice is called a trial. Typically, one observes that a particular alternative occurs increasingly frequently—this was called *learning*—until the system stabilizes so that no more average changes in behavior occur—this was called the completion of learning.

The concept of learning has been tested in several models, including living systems. In particular, experiments in mice have been used in order to test the learning process. The model proposed by Bush and Mosteller was applied to several sets of data. The first set included those reported by Stanley JC from seven rats in a T-maze experiment [51]. In each trial the rat could turn either left or right in the maze, and for the portion of Stanley's data being considered, the rat always found food on one side, and never found food on the

other side. The second set of data was obtained by Bush and Mosteller from five Harvard undergraduates operating a machine called the "two-armed bandit". On each trial, the subject pushed one of two buttons; one choice was always followed by a penny reward and the other side never led to reward. The third set of data was obtained by Bush RR, Davis RL and Thompson GL on six high school students in Santa Monica, California. In this experiment, the subjects were presented with two ordinary playing cards, face down, in each trial, and they were told to turn over one of the two cards; if the card turned over was a heart or diamond they received a reward of a nickel [43,52,53]. All cards in one position were reward cards, and all cards in the other position were non-reward cards. In all three experiments, there was a reward choice and a non-reward choice, which was elementary to the learning process.

### 3.1. Learning and Its Application in Information-Driven, Complex Artificial Life

Information theory has lately been the point of attention for researchers who are interested in the self-organization of artificial life or complex behavior. A concept of these approaches is derived from the notion that living organisms are information processing systems and in particular, knowledge processing systems. Recent research has also shown that the optimization of knowledge acquisition might be an evolutionary advantage. Although these ideas are quite intriguing, much interest is focused on the question how a general principle can be found for acquiring an artificial system with an internal drive for innovation or curiosity, or even more complicated with self-awareness. These approaches concern an artificial system that is characterized by a craving for increasing access to information about itself and the environment [54]. Sooner or later, a strategy for an open-ended, self-determined development of the artificial system will emerge [54].

The use of a proper measure for information was directly made clear by the development of information-theoretic approaches. Maximum randomness processes, like noise, could be favored by the attempt to maximize Shannon's information, making it, therefore, an indirectly feasible choice. Nevertheless, taking into account the fact that the behavior of an artificial system can be as random as possible, this would be an optimal solution. Alternatively, the Kolmogorov complexity is a factor that could be used. Schmidhuber's approach to artificial curiosity and self-motivation was based on this factor. In addition, a set of a univariate and multivariate statistical set was introduced by Lungarella et al. (2005), for the information structure quantification in motor and sensory channels [54,55]. In terms of emphasis, the generic criteria of information theory could vary. The different forms of focus, appear in the following examples: on maximization of empowerment; lack of homogeneity reduction over multiple agents' states, evaluated either with Boltzmann entropy states of swarm-bots, or with Shannon entropy of rule-space difference; spatiotemporal coordination increases into a modular intelligent robot, evaluated by the exceeding entropy calculated through a variable time series states of modules, and so on. All these examples of self-organization driven by information is the appraisal of the perception-action loop (or sensorimotor) with regard to information theory.

For example, the amount of Shannon information from the environment that one agent is able to inject within its sensors through actions completion, influencing future perceptions and actions, is evaluated by empowerment. In technical terms, for the behavior of a predetermined agent, the definition of the empowerment is stated as the agent's motivation channel capacity: the channel's max Mutual Information (MI), over the host of potential distributions of actions (i.e., the transmitted signal). At the same time, the exceeding entropy's maximization throughout a time period, stated in Prokopenko et al. (2006), permits the change of logic of controllers (i.e., agent's behavior changing) into a modular robot in a manner that manages to coordinate its actuators [54,56].

In nature, autonomy is an ambiguous phenomenon, as well as a significant dare in the artifacts world. One of the main characteristics of autonomy for both artificial and natural systems is noticed in the ability of autonomous investigation [57]. In the human and animal world, the capability to modify their own way of operation apart from a required

feature for survival and adaptation into conditions, it also offers a knowledge system with new information in order to improve its intellectual efficiency and development. Effective investigation in high-dimensional spaces is a significant challenge in the creation of learning systems. The well-known investigation-exploitation reciprocity was thoroughly studied concerning the field of augmented learning. In a Bayesian expression this reciprocity can be ideally solved, although it is untamed in computation. An even more dedicated solution to this concept is to deliver the agent with an internal stimulus for concentrating on specific stuff and, therefore, limiting the space of investigation. To address this matter in an even more basic way, we contemplate procedures for goal-free investigation of a physical system's dynamic properties as, for example, a robot. If the investigation is embedded in the agent by a self-identified manner, i.e., mostly as a deterministic function of variables regarding the internal status despite, through a pseudo-random generator, having the opportunity to elude the imprecation of dimensionality [57]. The reason is because certain particularities of the system, as for example restrictions and other implementation effects, are probably utilized in order to minimize the space for exploration. Therefore, a strategy of investigation that takes into account the specific environment and body is essential for creating capable learning algorithms for robotic systems of high-dimension. The question is how goal-free exploration can be beneficial to finally aim at goals?

In 2013, Martius et al. (2013) attempted to indicate that sort of coordinated patterns of a sensorimotor are developed that could be used for a quick construction of behaviors with extra complexity by using learning in a second level. Also, in a more direct view they can be combined with augmented learning where the standard adventitious investigation is replaced or increased by the goal-free investigation resulting probably in a considerable acceleration [57].

This is a general problem, the solution of which requires a basic example to be relative for a sizeable category of systems. Nowadays, information theory is at the cutting edge of the research, focusing on a group of relevant issues extending from evaluating and perceived to be autonomous systems even more capable of understanding problems of how the robot behavior's self-organization is related with naturalness and inspiration in technical systems and biology [57].

Systematically approaching the problem, both a lusty real-time algorithm in order to maximize the information measure and a suitable circumscription of that measure are required. The entire information of the experience from the past that can be used is quantified by the prognostic information that results from a process in order to predict forthcoming occurrences. From a technical view, in terms of time series, it can be determined as the reciprocal information, which ranges between the past and the future. It has been contended that prognostic information, also defined as transgression entropy and efficient measure intricacy, is the most physical complexity standard regarding the time series. By default, prognostic information that comes from sensor action is high when the artificial life succeeds in producing a flow of sensor values containing a large amount of information (in the sense defined by Shannon) by using practices leading to prospective outcomes. Consequently, by maximizing PI in an artificial life, the expectation is to show a high diversity of behavior and at the same time avoiding becoming purely random or chaotic. In the current status, somewhere between chaos and order, the artificial life will investigate the spectrum of its behavior in a way determined by itself under the sense already discussed above. From these rules a definition of a mechanism results for the variability in behavior as a deterministic function which is created at the synaptic level. Regarding the linear systems, various features of the method for the PI maximization have been presented. Specifically, it could be proven that the principle cause the behavior space of the system to be explored in an orderly way. In a particular case with a stochastic oscillator system, the PI maximization caused the system's controller to sweep through the area of given frequencies. More importantly, if a latent oscillation is hosted by the world, the PI maximization will train the controller to result in a resonance with this innate form of the world. This is emboldening, since at least in this simple example, the meaning of maximiz-

ing the PI is to strengthen and recognize the latent forms of the robotic system. Common measures of information theory are improved in the steady state. For a robot, this is not sufficient regarding the behavioral development of self-defined processing. In addition, as far as the robotics are concerned, the application of measures of information theory is often confined to the occasion of an action space of definite state with distinguished sensor values and actions. In addition, these constraints are transcend in this manuscript so it can be immediately used in physical robots with state and action space of high dimension. Unlike to the linear case, an amount of new phenomena are introduced by the non-stationarity and the nonlinearities. For example, in a simple hysteresis system the self-switching dynamics as well as the unprompted coaction of systems are coupled in a physical way. In systems of high dimension one can observe reduced dimensionality patterns of behavior which are dependent on the body as well as on the robot's environment.

### 3.2. Information Theoretical Concepts in Artificial Life

Nowadays, the researchers show an intensive interest regarding the question of finding generic mechanisms which support systems such as artificial life to obtain more autonomy. The approximations are broadly scattered, following several paths conforming with the certain categorization, as given in the work by Martius et al. (2013). Recently, information theory has been used in a group of approximations in robotics, to (i) perceive the way that the behavior can structure the input information, and (ii) proceed to the quantification of information nature that flows within the brain, as well as in a behaving robot [57]. Empowerment is considered as an interesting measure regarding the information, which quantifies the Shannon's information amount that can be "injected within" its sensor by an agent through the environment, in a way that affects future perceptions and actions. Lately, it has been proven that empowerment is a sustainable purpose for the self-defined behavior development in the problem of pole balancing and more agents regarding continuous domains. The exploration driven with PI maximization could also be considered as an alternative option to the homeokinesis principle since it has been successfully applied to a wide range of complex robotic systems. In addition, the aforementioned principle has been extended in order to create the core for self-organized and guided behavior.

The self-directed and self-defined exploration for embedded and autonomous agents is intimately linked with many latest attempts to provide the artificial life with a moving system that produces internal re-compensation signals for learning amplification in pre-determined tasks. An innovative work of Schmidhuber uses the progress of prediction as a re-compensation signal so as to inspire in the robot curiosity for new experiences [57]. The "playground experiment" includes related ideas that have been proposed. Also a few proposals formed by autonomy a hierarchy of capacities using the prognostic error of skills models or in a more abstract way to balance challenges and skills. Moreover, prognostic information can be used as an inherent stimulus in learning enhancement or added appropriateness regarding the evolutionary robotics. This approximation brings in further physical activities of the artificial life allowing an effective exploitation of embodiment outcomes. Self-determined is implied as "based totally on its intrinsic laws". In the animal kingdom, there is a growing substantiation which shows that the chain of animals from spineless creatures to mammals (including fish and birds) are endowed with an astonishing degree of diversity in response to a stimulation from the external environment. Hitherto, the reason for the creation of the particular variance in behavior has not been clarified. At the molecular level, the ideas are encompassed in the entire range from thermal fluctuations to quantum effects to the pure spontaneity assumption, rooting the variance in the presence of purely deterministic and intrinsic processes. If the behavioral variance in animals is provided in an identical manner, this can provide new awareness regarding the free will enigma.

### 3.2.1. Predictive Information (PI)

Regarding PI, a substantial aim is for the approach to be made independent of any discrimination either of the state or/and the action space in order for it to be directly useful in the approach of dynamical to artificial systems. The entire information of a previous experience is quantified by the PI of a process and it can be used in future events prediction. From a technical point of view, it is determined as the MI between the past and the future. It has been disputed that PI, also called effective measure complexity and excess entropy, is considered as the most natural intricacy rate regarding the time series.

The behaviors originated by the PI maximization are characterized by a high PI if the artificial life (due to its behavior) succeeds in producing a flow of sensory values containing high information (according the sense stated by Shannon) under the containment, although the results of the robot actions can be still considered as predictable. Consequently, the expectation by maximizing the PI of the artificial life is to show a wide variance of behaviors but without getting purely random or chaotic. For this regime of work, covering the distance from order to chaos, somewhere in between might be expected from the artificial life to investigate its behavioral potentialities in a most efficient manner. The exact reason that this works is made clear in the specific dynamical system, which is analyzed below.

### 3.2.2. Predictive Information and Dynamical Systems

The application of measures in dynamical systems of artificial life regarding information theory, in the vast majority is limited to the case of a space with finite state action and conspicuous sensor and actions values. Concerning the artificial dynamical systems, during the last 20 years the advent of a new tendency of control has been studied, which is penetrated more deeply in the approach of dynamical systems, using action and sensor variables in a continuous manner. This particular approach is very attractive since it allows the effects of the embodiment to be exploited in a more efficient way and renders more natural motion regarding artificial life. For example, many prosperous realizations of the "morphological computation" are accomplished by the use of recurrent neural networks which behave as the controller of the (dynamical) system consisting of the body, the brain, and the environment [54]. The approach of information theory regarding the representation of the dynamical systems needs still to be worked out in more detail. Moreover, this is the major motivation of this article, in order to present some initial results in this particular direction.

Zahedi et al. (2010), suggested a generic learning norm which has been extracted from the PI in a space of specific state and action, by using the technique of natural gradient [54,58]. The approach, from the information theory view, can be considered also as an alternate to the homeokinesis principle as reported by Der and Liebscher (2002); a systematic approach to the behavioral self-organization which has been successfully applied in a wide number of intricate robotic systems [54,57,59]. In addition, the above principle has been extended also in order to constitute a basis for self-organized behavior [54,57,59].

Can an artificial system develop its skills totally by itself, guided by the single objective to obtain even more information regarding its body as well as its intercommunication with the world? The above question immediately generates further questions: (i) what is the proper information related with the artificial system, and (ii) how can one find a suitable learning norm that fulfills the gradual ascent on the measure of this information? However, in a linear world, there are already various effects that establish the value related with the principle of information maximization. Specifically, it has been presented that the noise characterized as anisotropic impels in a systematic way the system to look into its behavior space. In this case, the maximization of PI caused the sweeping of a stochastic oscillator system's controller via the space of existing frequencies.

Above all else, if the world of the controller with which hosts are interacting in a latent oscillation, then by PI maximization the controller will memorize in order to become resonant with this particular internal manner of the world. This may considered as encouraging, because at least in a simple example as this one, the meaning of maximizing

the PI is the amplification and recognition of the robotic system's latent modes. In a way, by maximizing the PI, the artificial system will obtain the capability in detection of its physical prospects. Regarding the exception of isotropic noise, the principle of PI maximization conducts simple learning rules in which an entirely local formulation can be given. Actually, the standard back-propagation is only needed jointly with a Hebbian learning step [60–63]. Performing any operations that are non-local or sampling is not necessary. Indeed, this is a result of a system's linearity as well as the noise's isotropy. Nevertheless, the results of a report containing non-linear systems showed that an equivalent structure can also be obtained in the prevalent case [54]. This may be helpful in covering the gap by throwing bridges across the standard realizations of artificial neural network (the one with supervised learning) which present success in artificial life and the methods of information theory which hitherto have been placed on the discretization basis and burdened with implicated learning rules and high sampling efforts.

At that point, the question that arises is whether an artificial system could develop its capacity in order to achieve more self-consciousness on its own, guided by the sole purpose to collect information regarding the limits of its entity and intercommunications with its environment. This question raises at once further questions such as: (i) what is the artificial system's relevant information, and (ii) in which way can one find a suitable learning rule that accomplishes the gradient ascent on a measure of this information? Commonly, in the linear world, various examples exist that show the PI maximization principle's value. Specifically, the system explores its behavior space driven by the anisotropic noise. The PI maximization in this case sweeps the controller via the available frequencies' space adjusting its behavior. The PI maximization implies the increase and identification of the robotic system's latent modes. This is thought-provoking, because if a latent oscillation is hosted by the world's controller, in that case it will memorize to reciprocate using this particular intrinsic mode. As a result, in an artificial system the maximization of PI provides to some extent the ability to detect possibilities and body limits. During the existence of isotropic noise, the principle of PI maximization, urge simple and clear learning rules, in which a local formulation can be given. In fact, a step of Hebbian learning along with standard backpropagation is what is needed. The examination or the implementation of any non-local tasks is not a requirement. Apparently, this is an outcome of the system as well as of the noise's isotropy. In any case, the results of non-linear systems indicated that a potentially able to be compared structure can be achieved further in the general case as well as in approximations in any event [64]. This could fulfill the need to overcome the obstruction to standard neural network (with supervised learning) achievements. These are quite efficient concerning the artificial life and the techniques of data theory which depend so far on discretization and are getting in trouble with high examining attempts and the inclusive learning rules.

*3.3. Neuronal Systems as Forms of Artificial Life*

Once the brain interacts with the environment, it adapts in a constant manner and the form in which the environment is represented is entitled an "internal model". The dynamical properties and the connectivity of neurons are the expression for internal models' neurobiological basis. Consequently, the intercommunications between external devices and neural tissues present a fundamental means for the connectivity investigation and the dynamical properties of neural populations. Valentino Braitenberg developed and suggested this idea in the 1980s in order to investigate and represent the neuronal populations' dynamical behavior of in the lamprey's brainstem [65]. The maintenance of the brainstem took place in vitro. Two types of artificial device were used for its connection in a closed loop: (a) a small mobile robot and (b) a simulated dynamical system. In both cases, the recorded extracellular signals control the device and its output was interpreted in a form of electrical stimuli, which transferred to the neural system. The objective of the initial study was to evaluate the dynamical dissociation in the preparation of neurons in a configuration in the form single-input/single-output. The dynamical dissociation refers to the number

of variables of the state that determine the system's output along with the applied input. Recent report's results pointed out that whilst the particular neural system has considerable dynamical properties, its efficient intricacy, as set up by the dynamical dissociation, is rather temperate. In another study, a more specific situation has been considered, where a robotic device is controlled by the same section of the nervous system in a configuration of two-input/two-output. The input-output information from the neuro-robotic preparation has been adapted to neural network models with different interior dynamics, providing thus the ability to observe for each model its generalization error. In the brain–machine interface context, a computational and experimental framework such as this equips the means in order to investigate neural plasticity as well as internal representation.

In neuroscience, information transfer has been the topic of intensive investigation. In its simplest form, neuronal processing consists of an input signal, a processing unit (which is actually the neuron itself) and an output, which is the occurring behavior (as described by Bush and Mosteller (1953)). Information flow takes place in sequences of symbols, which could correspond to trajectories of stochastic processes [66]. Early works on neuronal information transmission, have found that information is transmitted by "signal trains" of discrete action potentials [67]. The first models investigated were simple neuronal connections, where one input pulse produced one output signal. An interesting finding of early works was that the brain was characterized by highly irregular, inter-spike activities, indicating that the spike-signal was noisy [67]. Hence, similar early works attempted to reduce this irregularity by accounting for the average of multiple signals. Although, this reduced the noise it is now known that even slight irregularities in brain signals still carry out messages [68]. The same was hypothesized for molecular signals in the cellular environment, which has been found that the most important differences were those that took place in infinitesimal ranges. Finally, it is possible that irregularities in neural signaling may actually represent the actual information transmission [67,68]. Overall, neuroscience stated two hypotheses, about the nature of neural information processing. The first theory describes the neural code as a "temporal code", which takes into account the neural "spike trains" ("Spike trains are a representation of neural activity. In neurophysiological studies, spike trains are obtained by detecting intra- or extracellularly the action potentials, but preserving only the time instant at which they occur. By neglecting the stereotypical shape of an action potential, spike trains contain an abstraction of the neurophysiological recordings, preserving only the spike times" [69].) [70,71] and the second theory are known as "rate code" theory assuming that the neural code in included in the spikes frequency, defined by the spikes emitted per second [72]. These processes can be described by different mathematical models, such as Markov and Bernoulli processes. However, even in its simplest form (just one neuron) understanding the mechanism of information transmission still remains elusive [73,74].

These topics are used in the field of computational neuroscience, where neurons are dealt with as binary electrochemical switches [75]. Several works have suggested that a spiking neuron can be considered as a system with memory, under two stable conditions [76], that are excited and not excited. This property leads to the idea that neurons firing or resting can be described as binary codes with 1 and 0 states, respectively. This assumption leads, in turn, to the definition of neural activity as a Shannon entropy phenomenon described as:

$$H = p \log_2 \left( \frac{1}{p} \right) + (1 - p) \log_2 \left( \frac{1}{1 - p} \right) \tag{10}$$

where $H$ is the number of bits of Shannon entropy in an action potential, and $p$ is the action potential's initial probability [76]. Equation (10) is used by numerous studies in order to estimate Shannon's entropy in a binary probabilistic system. The most common method of information calculation for a neural spike is by dividing a signal into evenly distributed time intervals, where the excitation or resting can be assigned to 1 and 0 values. Another suggested method included the investigation of information in neural spikes

in a relation between the input and output signals that would be the stimulus and the response [56,66,73,76].

### 3.3.1. Neuro-Robotic Systems

The critical objective for the development of efficient interactions between artificial devices and the brain is to understand and control neural plasticity and neural dynamics. Over the past decades, various experiments have addressed in a direct way the nervous system's ability in internal models creation of the controlled dynamics. A joint element of these analyses is the establishment of an open interaction in both directions between an external system with dynamics able to be programmed and the biological controller. To understand the external system's dynamics is considered as a fundamental challenge regarding the prosthetic devices' control as well as the brain-machine interfaces' clinical application, which deemed an emerging technology having a considerable clinical potential. In late studies, the measured information of primates' motor cortex, seemed to be used for the robotic arms guidance or for the computer cursors movement [77]. Nevertheless, the nervous system training task for an artificial device control, is still discouraging. In robotic systems, the flexible illustration of a controlled beneficial load is performed typically by a programming language. Such a representation has a biological counterpart which is gathered in human brains by amendable (or even "plastic") models of neural excitability and connectivity. These models could be considered as the biological programming language's components, the rules of which are still mostly unknown. There are various studies where the nervous system of a lamprey has been studied in detail and in particular its capability to generate and modulate the behavior regarding locomotion [77]. A part of neural circuitry was selected which integrates sensory signals such as the vestibular and generates motor commands for the stabilization of the body orientation during swimming. The particular system has been shown an adaptation: one-sided infestations of the capsules contained in the vestibule are followed by a sluggish restructuring of the neuron's activities until the recovery of the proper postural control. This kind of adjustment is the first indication of an internal model and the second indication is the learning after effects, which is an event that is observed when the disturbance is withdrawn and a mistaken behavior is observed while the invalid conditions are restored. One-sided infestation is an inconvertible operation and so the after effects' observation is improbable. However, this is not the hybrid systems case, in which the light sensors' sensitivity can be disabled and then after adaptation is again enabled to observe the adaptation's after effects.

### 3.3.2. A Closed-Loop Brain/Machine Interface (BMI) for Estimating Neural Dynamics

The interaction between external devices and a neural population is considered as a closed loop. This has the ability to be exploited in order to extract a short element of general dynamical properties contained in the neural population. Latest studies that observed the hybrid systems' behaviors have been used to estimate the neuron's dynamical dimension in a reticular formation of lampreys. The dynamical dimension is defined as the number of variables of independent state determined at each instant, the output of a system. Mathematically speaking, the following state and output equations describe the connection of the artificial system to the neural tissue [65]. In the same way the neural preparation can be described as a dynamical system. The main hypothesis of this study was that *there exists a state representation (s) of the neural preparation such that the changes of state are completely determined by the state itself and by the input to the neural preparation.*

### 3.3.3. Recurrent Dynamics in a Neuro-Artificial System

Experimental results concerning the interaction of brain neural cells with a machine have been described by Karniel et al. (2005). In their work, a part of the brainstem of the *Petromyzon marinus* (sea lamprey) in its larval state, was considered as the experimental setup of the hybrid system's neural component. This included the larvae anesthesia, using

tricaine methanesulphonate, the dissection and maintenance of the whole brain obtained in Ringer's solution in order to be continuously oxygenated, refrigerated and superfused [77].

The synapse vestibular/reticulospinal has been chosen, in their work, for the BMI for the following reasons: (a) it was comparatively well understood in regard to its physiology and anatomy; (b) it permitted access to populations of neurons under visual control; and (c) the entire brain could certainly be retained in vitro by plunging it in low-temperature Ringer's solution. The neurons' activity was recorded extracellularly in an area of the reticulated form, a relay which connects the vestibular, visual and tactile sensory systems, as well as central commands to the spinal cord's locomotor centers. In the Posterior Rhombencephalic Reticular Nucleus (PRRN) axons, a recording electrode has been placed.

In addition, among the Posterior Octavomotor Nucleus (nOMP) axons a unipolar electrode for stimulation of tungsten was placed. The stimulating electrode was placed on the side of the line in the middle and opposite to the electrode that records the signal is the point where the stimulating electrode is situated. The stimulating electrode was placed close to the nOMP, stimulating thus an extensive amount of fibers crossing the midline. This affected principally stimulant responses in the neurons that follow. A data acquisition card acquired the recorded signals at 10 kHz. The parameter of duration of maximum spike was set at 1 msec and the parameter of magnitude of least spike was set to 1.1 mV. To avoid any potential confusion among spikes and artifacts related with stimulus, the raw signal that acquired directly following each pulse/stimulus was refused. The duration of refused signal (the period of artifact annulment) was set to 3 msec. Their work has shown that neuro-robotic systems provide an environment for studying the operation of the nervous system.

### 3.4. Challenges in Neuro-Artificial Systems

Brain–machine interfaces are frequently investigated focusing on the machine to be used as tool for the disabled assistance. However, they can also be considered as hybrids of artificial life, where the traits of a biological system are exploited and connected to a machine-like instrumentation. To develop such tools, a necessity has arisen to gain a neural behavior comprehension from its operational aspect. The use of term "operational" hitherto, aims to distinguish the benefits by the signal behavior comprehension from the benefits of comprehension of the molecular and cellular underlying mechanisms. In the aforementioned experimental paradigms, a lamprey's brainstem was set in communication with both physical artificial devices as well as with simulated ones. In the first experiment, the simulated devices utilization allowed the estimation of the neural dynamics complexity, in terms of dimensionality in regard to its state and space. An advantageous feature of simulated devices similar to those used here, is the ability to establish arbitrary and well-structured dynamic properties. This provides the ability on the one side to design the simulated system for the excitement of considerably wide dynamical ranges and also on the other side to combine the neural tissue with different dimensions' artificial systems in order to test the evaluated neural dynamics' stability. The result of this initial study reveals the significance of the dynamical behavior even in a neural system of this particular simplicity, which between recording electrodes and stimulation consists of a single neurons layer. Nevertheless, the rather limited dimensionality which characterizes a behavior of single input/single output should be mentioned, with the range to be estimated between the values 2 and 4. In other reports, methods have been developed for the separation of noise from chaotic dynamics, based on the progressive infusion of the artificial noise in enhancing quantities on the data which is under analysis [78,79]. It is important to highlight that recognition of the dynamics of a neural system is based on an external device interaction in a two way manner. This can be performed in conjunction with a diversity of methods for the non-linear dynamics exploration, but is surely not limited to the described approach. Regarding the second aforementioned experiment, a neuro/robotic hybrid system was used in order to examine the neural population properties driving a robot based on two wheels. Given the conjunctive observations of neural responses and of robot

motions, it was deduced that a linear and dynamic network with repetitive connections, and on the same side of the body, is considered as the ideal model of the neural element operation. One account ascribes the repetitive dynamics to a realistic neural route. The potentiality for the contrary side of the body routes has been examined for pathways to the back and spinal cord. Nevertheless, it was noticed that a spinal cord surgical cross section has not shown any significant influence to the perceived dynamical behavior. One explanation doubtlessly recommends the existence of a local memory system enunciated by essential neural properties [77]. The particular system could be a mechanism of any form, as in particular the gate for plateau contingent, competent of inducting a relationship among the trend of a neuron to kindle at one moment and the condition of the neuron until a couple tenths of seconds before. The important matter is the fact that the neural operation is not totally related to the synaptic input of a certain moment. The neural routes that were switched on contained principally vestibular afferents, despite the existence of visual and cutaneous routes as well. From a standpoint of information processing, there is an essential equivalence degree between vestibular input and the optical switch which is generated by the sensors of robot. Both of them are right-left mechanisms and a phototaxis of positive sign matches the vertical direction tracking. The stimulus of semantics (gravity per light) is quite impossible to play an essential role in the present example. Hybrid neurorobotic systems offer a non-physical environment (in the sense of artificial) that is subject to reversible and controllable disturbances for examining the nervous system's activity. In the other experiment, by changing the light sensors' output gain a "reversible artificial lesion" it has been introduced. This process was attained as an alternative to irrevocable surgical management, as for example the unilateral labyrinthectomy which is the extraction of a vestibular organ. Compared with the actual lesion, the artificial lesion provides an explicit benefit which is its total reversibility. Zelenin et al. (2000) presented another study for this kind of neurorobotic interfaces [77,80]. By contrast with our study in which a direct excitation was used, this study presents an electrical motor which was used for the lamprey rotation so as to give feedback by the lamprey's natural sensory system. Neuro-robotic systems offer several significant advantages for the researcher as well as for the investigation of information transmission. This is achieved through the feedback that is provided from the artificial system and its sensors. The main idea behind the advantages of neuro-robotic systems, is that by changing the input-output and feedback mechanisms, it is possible to further study the function and information transmission in neural circuits. Examples of such mechanisms that need to be elucidated are the manifested plasticity of neurons, the connection between presynaptic input and post-synaptic activity, and others. The understanding of neural mechanics, could provide effective methods for "re-programming" neurons so that it can execute a desired task (whether this is desirable, is another question). This could be a central question for future research, since it could lead to the design and implementation of effective neural prostheses (or neural conscience re-programming?). Neural plasticity is probably the most important question that neuro-robotics could address. This could be a solid basis for establishing a working interaction between the nervous system and external devices. Neuro-robotic interfaces provide a new instrument for the direct investigation of how plasticity can be harnessed for generating desired behaviors.

## 4. Discussion

One of the most interesting questions stated in previous works was: what makes a particle a particle (in terms of quantum physics) and a gene a gene [3]. Based on previous theories, what defines those entities is information. Nonetheless, this may be true for particles, but for genes it is not fully applicable. Genes are considered to carry information based on the type of protein they can be translated to, yet genes express a tremendous variety of RNA molecules ranging from mature mRNAs to circular RNAs which, however, do not translate to a protein they carry a significant amount of information. This difficulty in defining "what makes a gene, a gene" brought about the concept of "individualization"

that is treating each biological entity as unique [3]. However, this could be considered as correct per se, some argumentation could be proposed. It is true that biological systems are so diverse in such a way that if one measures the expression of one gene in one specific tissue in one biological subject and the same gene in the exact same tissue in another biological subject these will be found to be different. How is it possible for the same gene to be different in two exactly similar situations? The answer could be that it is not the individual gene's effect in a biological phenomenon but the network of genes that make the difference. In that concept, information and thermodynamics come into play.

As mentioned, biological systems are open thermodynamic systems, exchanging energy with the environment in the form of heat, as well as mass in terms of nutrition, signaling, endocytosis, exocytosis etc. From the point of view of classical mechanics, all processes are deterministic with the exception of asymmetries due to the second law of thermodynamics and statistical mechanics, while biological phenomena could be described by *informational mechanics* [3,81]. In order for this to happen (that is to understand biological mechanics through information) it is essential to set a framework that considers the dynamics between the biological entities and their environment. Ideally, this could approach determinism if this process could be described by the laws of physics. As a solution to this approach it has been proposed that computation could serve as an alternative for investigating the interconnection between issues of information, thermodynamics and life [3,81,82].

## 5. Conclusions

In the present work we have reviewed and reported on the context of information and its connection to living organisms as well as its applications in neuro-artificial systems. These topics are of great importance, and have endless applications. The autonomous dynamical systems require an understanding of the transmission of information in biological systems. This process is largely unknown, and its basic mechanics, still remain to be elucidated. Towards this direction, the application of stochastic processes is expected to play an important role, since the basic biological mechanisms are of a stochastic nature.

**Author Contributions:** G.I.L. Conceptualization, methodology, formal analysis, investigation, data curation, writing—original draft preparation, visualization, resources, supervision; A.Z. Formal analysis, data curation, writing—original draft preparation; P.I. Writing—original draft preparation; D.K. Writing—review and editing, supervision, project administration. All authors have read and agreed to the published version of the manuscript.

**Funding:** This research received no external funding.

**Institutional Review Board Statement:** Not applicable.

**Informed Consent Statement:** Not applicable.

**Conflicts of Interest:** The authors declare no conflict of interest.

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
