# Peer review of "Information, Thermodynamics and Life: A Narrative Review"

_applsci, doi:10.3390/app11093897_

Round 1

Reviewer 1 Report

March 2, 2021

Review of the paper (applsci-1127400-peer-review-v1)

Information, Thermodynamics and Life: A Narrative Review 
by  George I. Lambrou, Apostolos Zaravinos, Penelope Ioannidou and Dimitrios Koutsouris

The idea of this paper is to link information with life to put some insight into the understanding of the brain. The Authors aim to show the connection between thermodynamics and information taking into account energy efficiency criteria. Although the manuscript deals with issues that have been strictly mathematically described, it does not discuss the theoretical background (mathematical descriptions), quantifying methods for this problem, or any technical solutions.
I find the idea presented in the paper interesting, but I have the following comments and concerns:  

A. There is a lack of mathematical descriptions of the information.

B. In my opinion, some brief paragraph about the fundamental problems in Neurosciences, specifically the problems concerning neuronal communications, the form of neuronal codes should be useful. Here are a few fundamental and recent papers that could be useful for the preparation of such a brief analysis:

- van Hemmen, J. L.; Sejnowski, T., 23 Problems in Systems Neurosciences, Oxford University Press, Oxford, 2006.
- Rieke, F.; Warland, D. D.; de Ruyter van Steveninck, R. R.; Bialek, W. Spikes: Exploring the Neural Code, MIT Press, 1997.
- Pregowska, A.; Szczepanski, J.; Wajnryb, E., Temporal code versus rate code for binary Information Sources. Neurocomputing 2016, 216, 756-762.  
- Pregowska, A.; Kaplan, E.; Szczepanski, J., How Far can Neural Correlations Reduce Uncertainty? Comparison of Information Transmission Rates for Markov and Bernoulli Processes. International Journal of Neural Systems 2019, 29 (8), 1950003-1-13.
- Mainen, Z. F.; Sejnowski, T. J., Reliability of spike timing in neocortical neurons. Science 1995, 268 (5216), 1503-1506.
- Street, S., Upper Limit on the Thermodynamic Information Content of an Action Potential, Front. Comput. Neurosci., 13 May 2020.
- Salinas, E.; Sejnowski, T. J., Correlated neuronal activity and the flow of neural information. Nature Reviews Neuroscience 2001, 2, 539-550.

C. The conclusion should be extended to stress/express the idea and analysis conducted in the manuscript.
The idea presented and developed in this paper is interesting and, in general, of high importance.  In fact, the paper should be submitted rather than a journal treating the philosophy of Science. In the case of considering publishing this article in Applied Sciences, given the comments above, I would recommend a major revision. The above concerns should be carefully addressed.

Author Response

Comments and Suggestions for Authors

March 2, 2021

Review of the paper (applsci-1127400-peer-review-v1)

Information, Thermodynamics and Life: A Narrative Review

By George I. Lambrou, Apostolos Zaravinos, Penelope Ioannidou and Dimitrios Koutsouris

The idea of this paper is to link information with life to put some insight into the understanding of the brain. The Authors aim to show the connection between thermodynamics and information taking into account energy efficiency criteria. Although the manuscript deals with issues that have been strictly mathematically described, it does not discuss the theoretical background (mathematical descriptions), quantifying methods for this problem, or any technical solutions. I find the idea presented in the paper interesting, but I have the following comments and concerns:

  1. There is a lack of mathematical descriptions of the information.

Response: We have amended the text as suggested (please refer to lines 146-158, 465-520)

  1. In my opinion, some brief paragraph about the fundamental problems in Neurosciences, specifically the problems concerning neuronal communications, the form of neuronal codes should be useful. Here are a few fundamental and recent papers that could be useful for the preparation of such a brief analysis: a) van Hemmen, J. L.; Sejnowski, T., 23 Problems in Systems Neurosciences, Oxford University Press, Oxford, 2006, b) Rieke, F.; Warland, D. D.; de Ruyter van Steveninck, R. R.; Bialek, W. Spikes: Exploring the Neural Code, MIT Press, 1997, c) Pregowska, A.; Szczepanski, J.; Wajnryb, E., Temporal code versus rate code for binary Information Sources. Neurocomputing 2016, 216, 756-762, d) Pregowska, A.; Kaplan, E.; Szczepanski, J., How Far can Neural Correlations Reduce Uncertainty? Comparison of Information Transmission Rates for Markov and Bernoulli Processes. International Journal of Neural Systems 2019, 29 (8), 1950003-1-13, e) Mainen, Z. F.; Sejnowski, T. J., Reliability of spike timing in neocortical neurons. Science 1995, 268 (5216), 1503-1506, f) Street, S., Upper Limit on the Thermodynamic Information Content of an Action Potential, Front. Comput. Neurosci., 13 May 2020, Salinas, E.; Sejnowski, T. J., Correlated neuronal activity and the flow of neural information. Nature Reviews Neuroscience 2001, 2, 539-550.

Response: We have amended the text as suggested, incorporating the suggested refences (please refer to lines 876-909)

  1. The conclusion should be extended to stress/express the idea and analysis conducted in the manuscript.

Response: We have expanded our manuscript with a “Discussion” section, where we added our comments for this interesting article (please refer to lines 51-56, 963-987).

The idea presented and developed in this paper is interesting and, in general, of high importance. In fact, the paper should be submitted rather than a journal treating the philosophy of Science. In the case of considering publishing this article in Applied Sciences, given the comments above, I would recommend a major revision. The above concerns should be carefully addressed.

Reviewer 2 Report

The paper is well written and has some merits but the discussion is missing key literature, one obvious one is related exactly to the topic of this paper, published by Entropy:

https://www.mdpi.com/1099-4300/14/11/2173

which the authors should consider discussing and contrasting with.

Author Response

Comments and Suggestions for Authors

The paper is well written and has some merits but the discussion is missing key literature, one obvious one is related exactly to the topic of this paper, published by Entropy: https://www.mdpi.com/1099-4300/14/11/2173 which the authors should consider discussing and contrasting with.

Submission Date

12 February 2021

Date of this review

20 Feb 2021 11:41:23

Response: We thank the reviewer for his/her evaluation. We have fully incorporated the suggested paper in our manuscript. We have expanded our manuscript with a “Discussion” section, where we added our comments for this interesting article (please refer to lines 51-56, 963-987)

Round 2

Reviewer 1 Report

The authors took into account all my suggestions and comments. I recommend publishing this article.